# Sleep Well and Recover Faster with Less Pain—A Narrative Review on Sleep in the Perioperative Period

**DOI:** 10.3390/jcm10092000

**Published:** 2021-05-07

**Authors:** Reetta M. Sipilä, Eija A. Kalso

**Affiliations:** 1Department of Anaesthesiology, Intensive Care and Pain Medicine, University of Helsinki and Helsinki University Hospital, 00029 Helsinki, Finland; eija.kalso@helsinki.fi; 2Sleep Well Research Programme, University of Helsinki, 00016 Helsinki, Finland; 3Department of Pharmacology, University of Helsinki, 00016 Helsinki, Finland

**Keywords:** sleep, insomnia, postsurgical pain, anxiety

## Abstract

Sleep disturbance, pain, and having a surgical procedure of some kind are all very likely to occur during the average lifespan. Postoperative pain continues to be a prevalent problem and growing evidence supports the association between pain and sleep disturbances. The bidirectional nature of sleep and pain is widely acknowledged. A decline in sleep quality adds a risk for the onset of pain and also exacerbates existing pain. The risk factors for developing insomnia and experiencing severe pain after surgery are quite similar. The main aim of this narrative review is to discuss why it is important to be aware of sleep disturbances both before and after surgery, to know how sleep disturbances should be assessed and monitored, and to understand how better sleep can be supported by both pharmacological and non-pharmacological interventions.

## 1. Introduction

### 1.1. What Do We Know about Interactions between Pain and Sleep?

Sleep disturbances and pain are both very common public health concerns. Approximately 30% of the adult population report some form of inadequate sleep [1] and about 8% report severe symptoms of insomnia [2]. About 20% of the European adult population report significant chronic pain [3], the percentage is higher (≥30%) in those over 70 years old [4]. Postoperative pain continues to be a prevalent problem as up to a quarter of patients undergoing operations report moderate to severe acute pain [5]. Risks of significant acute pain and its persistence vary between individuals and types of surgery, with, for example, more invasive surgery and preoperative pain in the surgical area causing more acute pain and increasing the risk of persistent pain [5,6,7].

Acute postsurgical pain differs both physiologically and psychologically from chronic pain, such as fibromyalgia or low back pain [8]. Physiologically, the tissue injury caused by the surgical procedure sets off a chain of inter-related events aiming to limit further damage, fight infection, and initiate healing [9]. Psychologically, the patient expects postsurgical pain and therefore worry, anxiety, and pain anticipation are inevitably part of the acute pain experience. Knowledge about factors relating to both pain experience and sleep disturbances reveals similarities in the development of both, such as negative expectations and psychological distress [10,11,12,13,14,15]. The physiological processes behind postsurgery healing may not be as efficient if the patient also suffers from insomnia [15,16]. This is discussed further below.

The bidirectional nature of sleep and pain is widely acknowledged. A decline in sleep quality increases the risk of pain onset and also exacerbates existing pain [11,17]. A large cohort study with more than 10,000 individuals found an association between different aspects of disturbed sleep (insomnia frequency, sleep onset latency, sleep duration, and sleep efficiency) and experimental pain sensitivity [18]. The associations remained significant even when the known comorbidities, psychological distress, gender, and age, were controlled for in the analyses.

Poor postoperative sleep, caused, for example, by anxiety and postoperative pain, may persist from days to weeks after surgery [19,20,21,22,23]. In practice, this may act as a trigger for the development of prolonged symptoms of insomnia. Acute sleep disturbances may cause exaggerated worry about not being able to sleep and thus further undermine sleep. The development of and the cognitive process behind persistent insomnia are discussed below.

### 1.2. Aim of the Present Review

This narrative review focuses on the associations between sleep and postoperative pain. Its main aim is to discuss the importance of being aware of sleep disturbances both before and after surgery, how sleep disturbances should be assessed and monitored, and how better sleep can be supported by both pharmacological and non-pharmacological interventions.

## 2. Perioperative Sleep Disturbances

### 2.1. Sleep Disturbances and Their Consequences

Occasional sleep disturbances are very common and familiar to most individuals. Insomnia, also called prolonged sleep disturbances, has been defined as a sleep difficulty that occurs at least three times per week and has lasted at least for one month. Symptoms of insomnia may include such aspects of poor sleep as difficulty initiating sleep, early awakening, lengthy or repeated awakenings during the night, and inadequate total sleep time [24,25], though not necessarily all of these. This definition of severe sleep disturbances also includes the effect of poor sleep on daytime functioning, such as cognitive functioning, alertness, and energy [24,25].

Prolonged sleep disturbance predisposes a person to mental disorders, such as depression, anxiety, psychosis, and alcohol abuse [26]. A wide variety of physical diseases have been recognized to be associated with insomnia, e.g., cardiovascular diseases [27], dementia and cognitive decline [28,29], and obesity [30]. A large cohort study found a three-fold increased mortality risk in individuals with short sleep duration and persistent symptoms of insomnia [31]. Another large cohort study showed that better quality of sleep was associated with better-perceived health and that pain was one of the most important variables to explain this association [32].

One potential factor that could explain the link between poor sleep and neurodegenerative processes is neuroinflammation [33]. An interesting new hypothesis is that glymphatic flow, most active during NREM sleep, clears the brain of proinflammatory waste products during sleep. Interventions that enhance NREM sleep could thus have both short- and long-term beneficial effects [34].

### 2.2. Sleep, Pain, and Anxiety—The Vicious Circle

Research has suggested that early-life stress exposure predisposes to changed activation of the hypothalamic–pituitary–adrenal (HPA) axis [35], which has been suggested to be accompanied by long-lasting modifications in stress reactivity [36]. Further, this might play an important role in vulnerability to hyperarousal reactions to negative life events in later life, and inadequate emotional reactivity as an adult, contributing to the development of chronic insomnia [35,36]. The involuntary reaction of the HPA axis forms the basis of a vicious circle of psychological disorders and insomnia.

The cognitive model of insomnia [12] was developed to describe the cognitive process behind the subjective symptoms of insomnia. It proposes that individuals suffering from insomnia tend to be excessively worried about their sleep and its daytime consequences. The negative cognitive activity triggers both emotional distress and autonomic arousal. It has been suggested that this triggers selective attention towards internal and external sleep-related threat cues, such as postoperative pain. The negative cognitive processing in pain and the symptoms of insomnia are similar. The worrying process, including pain-related fear and anxiety, maintains rumination and physical arousal [37,38]. In vulnerable patients, postsurgical pain may work as a trigger for the development of more severe symptoms of insomnia. Previous research provides a fairly solid background for the bidirectional association between mental health and symptoms of insomnia [26]. In a large longitudinal study of sleep, mental health problems were found to increase the risk of persistent insomnia nine-fold [39]. Figure 1 displays the association between pain, anxiety, and symptoms of insomnia.

### 2.3. Factors Associating with Postoperative Sleep Quality

Evolutionarily, it may have been beneficial to have lighter, more fragmentary sleep when an individual was wounded, and the postsurgical phase can be compared with that state. Wakefulness-promoting neuronal networks are easily activated by painful signals [40,41].

Sleep disturbances are quite frequent in patients in the postsurgical setting, and sleep patterns have been shown to be affected in several ways. Typical postoperative sleep disturbances include decreased total sleep time with several arousals, leading to decreased REM and non-REM sleep [10]. Patients may also report worse sleep quality and nightmares [42]. Sleep disturbances after surgery are likely to be caused by multiple factors, as described in more detail below.

The development of postsurgical insomnia is common in patients who were already suffering from poor sleep before surgery [43]. Greater age [42,44] and preoperative comorbidities, like obstructive sleep apnea [45] and coronary artery disease [46], have been found to systematically increase the risk of postoperative sleep disturbances.

### 2.4. Surgery-Related Causes of Sleep Disturbances

Various mechanisms, such as endocrine, autonomic, and inflammatory stress responses, have been suggested to explain the association between surgical trauma and sleep disturbance [42]. Some cytokines that are released after surgery are also known to affect sleep quality. Tumor necrosis factor-α (TNF-α), interleukin-1 (IL-1), and IL-6 are known to be associated with decreased REM sleep duration and with increased deep sleep (NREM, slow-wave sleep) duration [47,48]. High-stress hormone release, for instance, adrenocorticotropic hormones and cortisol, and sympathetic overactivity can also disrupt sleep [49]. Thus, it is understandable that more severe surgical trauma may be associated with greater sleep disturbance (e.g., open compared to laparoscopic cholecystectomy) [50].

### 2.5. Anesthesia- and Analgesia-Related Associations with Sleep Disturbances

Spinal and regional anesthesia seem to associate with a lower risk for postoperative sleep disturbances than general anesthesia [51]. This may be due to both a reduced stress response after regional or spinal anesthesia than with general anesthesia and a reduced need for anesthetic agents and opioids that may impair sleep. Higher postoperative consumption of opioids has been associated with more frequent sleep problems [51].

Studies in healthy volunteers have shown opioids to impair sleep architecture [52]. Opioids have been shown to reduce deep sleep, but increase stage 2 sleep, without affecting sleep efficiency or total sleep time [53]. NSAIDs have also been suggested to have a mild sleep-disturbing effect [54]. However, their anti-inflammatory effects and good analgesic efficacy in postoperative pain are likely to have a positive net effect on sleep. Corticosteroids are used as adjuvant analgesics and antiemetics in the perioperative period. However, little is known about the effects of sleep in this context.

The so-called adjuvant analgesics, such as gabapentinoids, ketamine, and dexmedetomidine, are interesting for their possible effects on postoperative sleep. Unfortunately, systematic reviews on the use of gabapentinoids or ketamine in perioperative analgesia have not assessed their effects on sleep [55,56]. A very recent RCT on perioperative ketamine concluded that most of the 189 patients in the study complained of disturbed sleep on the night following surgery. However, there were no significant differences between the ketamine and placebo groups [57], neither were there significant differences between the groups regarding hallucinations or confusion, and no patients reported vivid dreams or nightmares.

Elderly patients are particularly vulnerable to the CNS effects of anesthetic and analgesic agents. Postoperative delirium (POD) is a common complication in the elderly, with an incidence of 10–60% [58], and it is associated with increased morbidity and mortality. Sevoflurane, remifentanil, and fentanyl, for example, have been reported to exacerbate POD, while, interestingly, perioperative dexmedetomidine, an α_2_-adrenergic agent, has been shown to prevent POD by reducing the requirement of these drugs [59].

### 2.6. External Causes for Postsurgical Sleep Disturbances: Better Sleep Quality at Home

The first comprehensive study, from the Netherlands, surveyed the quality of sleep in the hospital wards compared with sleep variables at home. The results showed a few differences; in the hospital, the total sleep time was shorter, patients reported more nocturnal awakenings than usual, and they woke up much earlier than at home. Poorer sleep quality was mostly due to external causes that disturbed sleep, e.g., hospital staff visits in the rooms during the night, other patients, sound from the medical devices, lights from corridors and patients’ rooms, and nocturnal toilet visits [60]. Table 1 summarizes factors associated with poorer sleep quality after surgery.

## 3. Measuring Sleep

### 3.1. Objective Sleep Quality

Polysomnography performed in a sleep laboratory (or at home) is the ‘gold standard’ for the measurement of objective sleep quality. However, it is a quite complex and time-consuming procedure and it is used only to diagnose complicated sleep problems or for research purposes. Basic research on sleep is needed to understand how and why our interventions (pharmacological or other) influence sleep. Objective measures of sleep may help us to understand those structures of sleep that are unexpected and important interventional targets.

Different kinds of smart devices, developed in recent years, offer an easier but less precise way of objectively monitoring the macro-architecture of sleep [61,62]. Several consumer-oriented sleep trackers, e.g., Oura Ring [61] and Actigraphy [62], have been tested and found to be valid for monitoring different sleep-related variables, such as sleep efficiency and total sleep time [63,64]. A recent study used Fitbit trackers and sleep questionnaires in patients undergoing elective surgery and showed that they experienced severe inpatient sleep disturbances, worse than found in similarly studied ICU cohorts [65]. The latest version of some trackers also provides SpO_2_-measurement. These trackers are easy for patients to use and make sleep easy to measure in its natural environment. Trackers are convenient to use for the patients, particularly for long periods, unlike, for example, completing sleep diaries [66].

### 3.2. Subjective Measures of Sleep Quality: Questionnaires

Keeping a sleep diary is a widely used method of assessing subjective sleep measures, and its standardized use has been proposed [66,67]. Several questionnaires are also available to measure subjective sleep quality: the psychometric properties of the studied scales are quite good [68]. Table 2 displays some of the most widely used and validated questionnaires. The selection of measurement tools should be based on the purpose of their use (e.g., for daytime sleepiness or severity of insomnia). Like all methods used, sleep-related questionnaires should also be validated for the language of use. Most validated questionnaires have cut-off points that indicate the severity of the sleep difficulty or insomnia. They are mostly inexpensive, fast to complete, and feasible in clinical settings, while scoring is easy for the professionals. However, subjective sleep estimation by the patient likely also reflects the patient’s affective state [69]. Symptoms of anxiety and depression have been associated with the lower subjective estimation of sleep duration and poorer sleep quality compared with the objectively measured sleep quality [69,70,71]. It is important to recognize that the subjective estimation of sleep is also influenced by patients’ worries about the upcoming surgery and recovery. Thus, measurements of subjective and objective sleep quality give somewhat different information about a patient’s sleep health. Both aspects of sleep quality, objective and subjective, are important when screening sleep pre- and postoperatively.

## 4. Why Is It Important to Promote Good Sleep Quality after Surgery?

Poor quantity and quality of sleep can be harmful to postsurgical recovery [51,72]. In addition to the bidirectional relationship between poor sleep and pain, many comorbidities increase the risk for the development of postsurgical sleep disturbances, which can impair recovery, leading to longer hospital stays, functional limitations, and poorer emotional state and quality of life [73]. Poor quality of sleep can be considered as a mediating factor for poorer postoperative recovery. For example, patients with obstructive sleep apnea are at risk of developing symptoms of insomnia after surgery, further adding to the risk of POD [74]. POD, in general, is considered to be much more common in patients with sleep problems [75]. A study of cardiac patients who underwent percutaneous coronary intervention showed that disturbed sleep was an independent risk factor for cardiac events by the four-year follow-up [72]. This would suggest that high-risk patients having either primary (such as sleep apnea or restless legs) or other sleep disturbances should be referred for appropriate sleep therapy preferably before, or at least after surgery.

After total knee replacement, good quality sleep was found to associate with both better acute pain relief and better functioning at the three-month follow-up [76]. Good postoperative management of sleep has also been shown to reduce postoperative duration of hospital stay and sick leave [51].

### Quality of Sleep and Risk of Postoperative Acute Pain and Pain Persistence

Quite a few physiological, surgery-related, and psychosocial factors have been established as risk factors for acute and persistent postoperative pain, including preoperative chronic pain conditions, type of surgery, age, and mood [5,7,13,76,77,78,79,80,81,82,83]. The most important psychological variables found to associate with persistent postoperative pain are anxiety, symptoms of depression, psychological distress, and pain-related catastrophizing [84]. Psychological vulnerability factors, especially anxiety sensitivity, are also critical underlying contributors in the development of insomnia, as discussed earlier in this review.

Quality of sleep has not frequently been assessed in studies about postsurgical pain. However, in a recent systematic review, preoperatively measured sleep problems were the strongest predictor of poor control of acute postoperative pain [85]. The risk was over two-fold greater in patients with self-reported prolonged sleep difficulties.

A study in breast cancer patients found an association between poor sleep the night before the surgery and the severity of acute postoperative pain [86]. Our own study showed an association between preoperative subjective sleep difficulties and the development of postsurgical neuropathic pain in breast cancer survivors with surgical nerve injury [82]. Sleep difficulties were also associated with poorer mental health-related quality of life four to nine years after the index surgery in these patients [87]. The most recent study found preoperative sleep disturbances to be a systematic risk factor for different aspects of persistent pain, such as pain severity and impact in breast cancer patients [88].

## 5. How to Improve Sleep Pre- and Postoperatively

As major sleep problems have been shown to associate with both more severe acute and persistent postsurgery pain, interventions to improve sleep quality should already be considered in the preoperative period, when elective surgery is planned. Primary sleep disorders such as sleep apnea and restless legs should be diagnosed and treated. Also, treatment of the underlying conditions that themselves pose an independent risk for surgery and an additional risk by decreasing quality of sleep, e.g., asthma and cardiovascular diseases, should be optimized before surgery.

The need and choice of treatment of sleep problems should be based on adequate preoperative risk assessment. The most convenient way to assess the severity of sleep disturbances is to use validated, subjective sleep quality questionnaires (see Table 2).

### 5.1. Non-Pharmacological Treatment

Non-pharmacological management of sleep is the first-line treatment for sleep problems, but not all methods may be feasible within the time available [96]. The basic rules of sleep hygiene should be gone through in the preoperative visit with the patient. These could include, e.g., regular sleep times before the surgery (7–8 h), avoiding the use of electronic devices in bed, and optimal room temperature. In the preoperative evaluation, patients should be advised to maintain the regular circadian sleep–wake rhythm while at the hospital, if possible, and after discharge at home. Non-pharmacological techniques that are likely to improve sleep in both pre- and postoperative settings and which only need short practice include, for example, relaxation techniques, breathing exercises, and listening to music [96,97,98,99]. The availability of these techniques has significantly increased with the availability of internet-based programs.

If time allows and the sleep problem is severe, cognitive behavioral therapy for insomnia (CBT-I) is considered the first-line intervention [100,101]. CBT-I has been shown to be effective in both non-clinical populations and in chronic pain patients [100,102,103]. In recent years, research on the effectiveness of CBT-I in chronic insomnia has focused not only on subjective evaluation but also on physiological assessments. The completion of CBT-I resulting in the remission of insomnia is also associated with lower levels of C-reactive protein [104] and IL-6 and TNF expression by monocytes [105], indicating a decreased inflammatory reaction. Also, inflammation-involved gene transcripts were downregulated after CBT-I treatment [105].

Third-wave psychological interventions, i.e., acceptance and commitment therapy (ACT) programs, have proved effective in the management of insomnia in chronic pain patients [106]. ACT-based programs concentrate on increasing cognitive and emotional flexibility in one’s life. As far as the authors are aware, there are no studies of the effectiveness of ACT interventions for insomnia in pre- and postoperative settings—this would be worth studying.

The use of CBT-I needs practice and insight on how to change one’s thoughts about sleep. It also takes time for sleeping habits to change and for CBT-I to show effects. CBT-I programs last from six to eight weeks, roughly, so symptoms of insomnia should be screened in good time before elective surgery. The risk factors for the development of persistent pain and the symptoms of postoperative insomnia overlap to some extent [12,37,38]. Therefore, preoperative screening should cover the symptoms of depression, anxiety sensitivity, extensive worry, self-efficacy, and how likely the patient feels she/he can modify sleep and pain experience.

### 5.2. Pharmacological Treatment (Pre-, Peri-, and Postoperatively)

The need and choice of pharmacological treatment of sleep problems should be based on preoperative risk assessment. If the patient is already taking drugs to enhance sleep, this treatment should not be interrupted before surgery. One systematic review concluded, in general, that there is insufficient evidence to suggest that pharmacotherapy improves the quality of sleep in hospitalized patients [107].

In short-term use, such as before and after surgery, benzodiazepines can be used for their anxiolytic effects. However, they do not improve sleep architecture. Zolpidem is a short-acting benzodiazepine imidazopyridine that binds to GABA_A_ receptors. In a small study in patients having orthopedic surgery, zolpidem improved subjective sleep quality, but not sleep architecture [108].

Ideally, the sleep-enhancing drug should also reduce anxiety and pain. Gabapentinoids would be particularly interesting as they are anxiolytic and also known to improve sleep [109]. A recent review on perioperative use of gabapentinoids showed that they slightly reduced opioid consumption and were not significantly associated with delirium [55]. So far, no study has specifically assessed their effects on postoperative sleep. Exogenous melatonin can have a beneficial effect in insomnia disorders and circadian disorders of the delayed sleep phase type [110]. A systematic review on the effect of melatonin on postoperative sleep concluded that most studies reported a beneficial effect on sleep [111]. Melatonin, administered before and after breast cancer surgery, increased sleep efficiency measured with Actigraphy. However, there was no difference between the melatonin and placebo groups for postoperative pain [112].

The choice of perioperative anesthesia and analgesia may also improve postoperative sleep. In this respect, dexmedetomidine is particularly interesting. It has been shown to prevent POD by reducing the requirement of opioids. Recently, it was also shown to prevent POD in the elderly who were anesthetized with total intravenous anesthesia [113]. In addition, low-dose (0.1 mg/kg/h) dexmedetomidine improved subjective sleep quality and increased total sleep time and stage N2 sleep in elderly patients in the postoperative ICU [114]. The noradrenergic tone has been shown to reduce NREM sleep and glymphatic flow, which has been associated with neuroinflammation since the glymphatic system has a pivotal role in the removal of metabolic waste from the CNS [115,116]. Dexmedetomidine blocks noradrenaline release from the locus coeruleus (regulator of arousal and awake states), increases slow-wave activity (0.5 to 3.5 Hz) [117], and has been shown to enhance glymphatic clearance in rodents [118]. Thus, it might have beneficial effects via this mechanism in reducing neuroinflammation. In support of this hypothesis, Hu et al. [113] reported that the surgery-induced increase in IL-6 levels was lower in the patients receiving total intravenous anesthesia (TIVA) with dexmedetomidine compared with those having TIVA without dexmedetomidine.

## 6. Summary and Clinical Implications

Recent research clearly shows that preoperative sleep disturbances are a risk factor for a number of postoperative adversities, including pain. Figure 2 summarizes interventions that can be introduced to patients preoperatively, in the hospital, and postoperatively at home to improve sleep.

***What should be done preoperatively:*** Patients should be assessed for potential sleep problems and those at high risk for insomnia identified. Patients should be advised and educated on good sleep hygiene and how to improve sleep. Suitable interventions for pre- and postoperative use should be planned and introduced. Patients should be supported to find individually suitable interventions. Motivation to rehearse non-pharmacological interventions to support sleep (e.g., relaxation and breathing techniques) is crucial. The effectiveness of the interventions depends on how well the patient is engaged in training in new skills to calm her/himself down.

***What should be done at the hospital:*** Given the bidirectional relationship between sleep and pain, an adequate control of pain remains a priority to support sleep after surgery. Techniques to improve sleep should be continued in the hospital (e.g., relaxation techniques). Hospital staff should encourage patients to use the skills they learned during the postoperative period. Impacts of external distractions should also be considered. For instance, novel technological developments may enable the use of remote measurement tools to measure vital signs. Nocturnal check-ups could be done via webcams to avoid unnecessary noise and change of lighting levels in patient rooms in the ward. Patients should be given earplugs and eye masks to minimize distractions. If possible, patients recognized as having known risk factors for disturbed sleep should be offered private rooms. These are low-cost changes but could save much in recovery time and satisfaction of the patients.

The evidence so far emphasizes the important role of sleep in recovery in the postoperative setting. Evaluation of sleep quality (subjective or objective) should be included as a factor when postoperative outcomes are studied. Consumer-oriented sleep trackers are an easy way to monitor a large variety of sleep variables before and after surgery. Measuring subjective sleep quality with questionnaires is time-saving and easy to administer. Subjective sleep quality measures provide information about the affective states of the patients. The interventions that support sleep are likely to have a positive impact on both the psychological and physiological recovery of surgical patients. Finally, the attitudes of the health care providers should favor fostering good sleep after surgery.

## Figures and Tables

**Figure 1 jcm-10-02000-f001:**
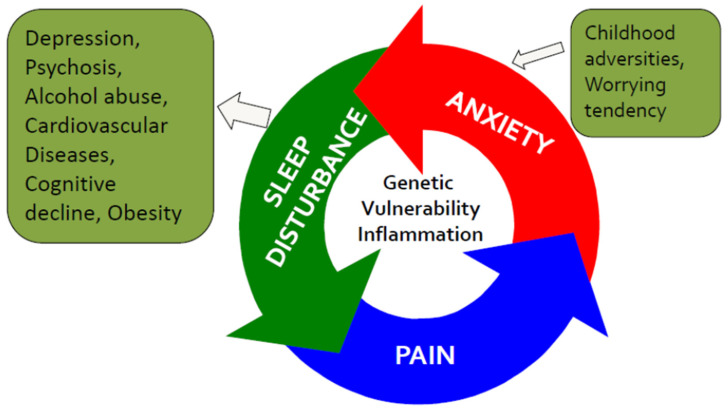
The vicious circle of the association between pain, anxiety, and disturbed sleep. Early-life exposure to stressful events is a vulnerability factor for anxiety, sleep disturbances, and pain in adulthood.

**Figure 2 jcm-10-02000-f002:**
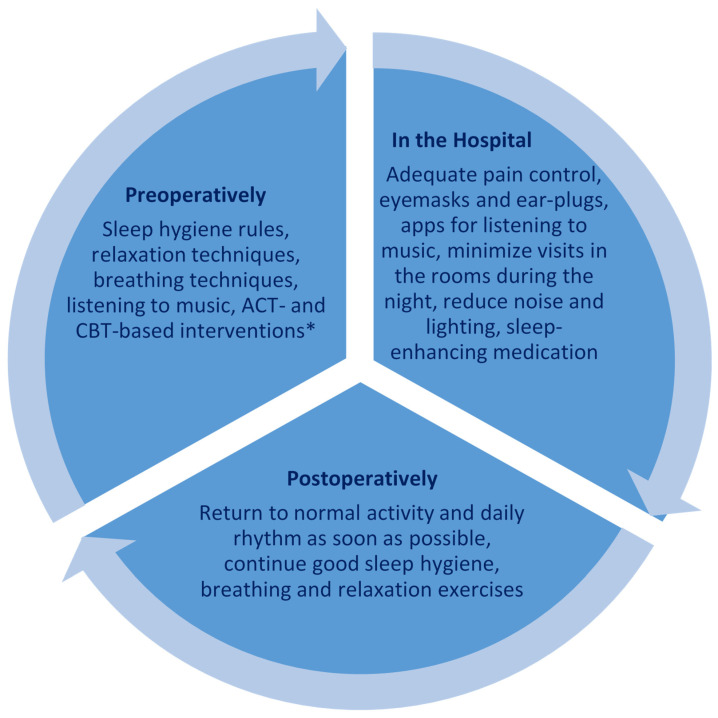
How to improve sleep quality pre- and postoperatively. Non-pharmacological interventions to support sleep preoperatively should be continued postoperatively. Abbreviations: CBT = Cognitive Behavioral Therapy; ACT = Acceptance and Commitment Therapy. * Need to be initiated about 6–8 weeks before surgery.

**Table 1 jcm-10-02000-t001:** Factors known to associate with poor postoperative sleep, and which are important to consider during the perioperative period.

Poor Subjective Sleep Quality Preoperatively
Symptoms of anxiety
Symptoms of depression
Surgical worry
Preoperative pain (surgical area or other chronic pain)
Severity of surgical trauma
Type of anesthesia (general anesthesia)
Type of postoperative analgesics (high dose of opioids)
External factors (e.g., light and noise in the ward)
Obstructive sleep apnea
Greater age
Coronary artery disease

**Table 2 jcm-10-02000-t002:** Examples of questionnaires to evaluate sleep quality pre- and postoperatively.

Questionnaires	Number of Items	
Sleep diary [66,67]	Consensus Sleep Diary (CSD) suggest to include 9 themes.	Assessed themes: Time of getting into bed, time at which the individual attempts to fall asleep, sleep onset latency, number of awakenings, duration of awakenings, time of final awakening, final rise time, perceived sleep quality, additional space for open-ended comments.
Insomnia Severity Index (ISI) [89]	7 Items	Assesses the severity of sleep onset, sleep maintenance difficulties (both nocturnal and early morning awakening), satisfaction with current sleep pattern, interference with daily functioning, impairment attributed to the sleep problem, and concern caused by the sleep problem.
Pittsburgh Sleep Quality Index (PSQI) [90]	24 Items	Assesses sleep quality, sleep latency, sleep duration, habitual sleep efficiency, sleep disturbance, use of sleep medications, and daytime disturbance.
Mini-Sleep Questionnaire (MSQ) [91,92]	10 Items	Assesses both symptoms of insomnia and excessive daytime sleepiness
Sleep Condition Indicator (SCI) [93]	8 Items	Assesses concerns about sleep quality, getting to sleep, remaining asleep, daytime functioning, daytime performance, duration of sleep problem, nights per week having a sleep problem, and extent troubled by poor sleep.
Epworth Sleepiness Scale (ESS) [94]	8 Items	Assesses the severity of daytime sleepiness, which is an important manifestation of sleep disorders.
The Richards-Campbell Sleep Questionnaire (RCSQ) [95]	6 Items	Assesses in-hospital sleep quality: sleep depth, sleep latency, awakenings, returning to sleep, sleep quality, and noise disturbance.

## Data Availability

Not applicable as this is a review article.

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
