# Peer review of "Sleep Well and Recover Faster with Less Pain—A Narrative Review on Sleep in the Perioperative Period"

_jcm, 2021, doi:10.3390/jcm10092000_

Round 1

Reviewer 1 Report

The review presented an interesting topic concerning the association between sleep disorders and pain. However, in my opinion, this paper does not provide more information than is generally known on this topic. The methods of diagnosis and treatment of patients with sleep disorders are presented - these information are obvious. This applies not only to patients with pain symptoms. The molecular basis of the effects of pain and sleep disorders has not been sufficiently presented. It is also worth mentioning about patients with chronic inflammatory diseases and chronic pain in this context. PSG is not always dedicated to sleep diagnostics, because it assesses a random night, and some sleep disorders concern a certain period of time, such as insomnia. In the treatment of sleep disorders, it is necessary in the described cases to treat the basis disease.  ​CBT is the gold standard for treating insomnia, but is it effective for a patient with severe pain? There is no discussion of the results, for example, "A study of patients who had undergone percutaneous coronary intervention showed a positive correlation between a greater incidence of sleep disturbances and an increased risk of cardiac events by the four-year follow-up" requires comment. What could be the reasons for this? The tables and figures do not add new to the topic and present generally known information for doctors in the field of sleep medicine.

Author Response

Thank you for the fast review of our MS entitled: “Sleep well and recover faster with less pain” and for the constructive comments.

Below, we answer each question (in bold italics). The changes in the revised manuscript are shown with “track changes”. In addition to the reviewers’ comments, we suggest that the title is revised to “Sleep well and recover faster with less pain – a narrative review on sleep in the perioperative period”

Looking forward to hearing from you,

Sincerely,

Reetta Sipilä, PhD

Review 1

The review presented an interesting topic concerning the association between sleep disorders and pain. However, in my opinion, this paper does not provide more information than is generally known on this topic. The methods of diagnosis and treatment of patients with sleep disorders are presented - these information are obvious. This applies not only to patients with pain symptoms.

The molecular basis of the effects of pain and sleep disorders has not been sufficiently presented. It is also worth mentioning about patients with chronic inflammatory diseases and chronic pain in this context                                                                                                                                             Thank you for this comment. However, discussing the molecular basis of pain and sleep disorders is a rather extensive topic and not exactly in the focus of this clinical review. We have added text on pain and inflammation, including comments on NSAIDs and corticosteroids, to p. 4, lines 141-144. However, we do not address chronic inflammatory diseases or chronic pain in general, in this review.

PSG is not always dedicated to sleep diagnostics, because it assesses a random night, and some sleep disorders concern a certain period of time, such as insomnia.                                                     We only mention PSG as a gold standard for most advanced sleep diagnostics. We have clarified this in the text. However, as diagnosis of more complicated sleep problems is not in the focus of our review we do not think that we should describe PSG and issues related to it in more detail.

In the treatment of sleep disorders, it is necessary in the described cases to treat the basis disease.  Thank you for this comment. We have now added text to address this on p.,6  lines 219-223 and 252-256.

CBT is the gold standard for treating insomnia, but is it effective for a patient with severe pain?       We propose under 5.1 that CBT-based sleep interventions are used in the preoperative period of elective surgery, if time allows. During this period the patients are unlikely to have severe pain. CBT is one of the most commonly used psychological interventions for chronic pain.  A very recent review also found a positive effect of iCBT in chronic pain patients. This is also mentioned in the text (Reference number 100, p.8, line 285-).

There is no discussion of the results, for example, "A study of patients who had undergone percutaneous coronary intervention showed a positive correlation between a greater incidence of sleep disturbances and an increased risk of cardiac events by the four-year follow-up" requires comment. What could be the reasons for this?                                                                                Thank you for this excellent comment. We have now added text to clarify this on p. 6, lines 219-223.

The tables and figures do not add new to the topic and present generally known information for doctors in the field of sleep medicine.                                                                                                 You are right, we agree. However, this review has not been written for experts in sleep medicine but for anaesthesiologists as this review is, if accepted, part of a special section for anaesthesiologists.

Reviewer 2 Report

This is an interesting review on benefits of sleep on postoperative pain recovery. However, the authors left some gaps and space between the information which made it difficult to grasp the topic most of the times. I would suggest that author revised this manuscript as suggested below to improve the comprehension.

  1. The section 2 heading says “Preoperative sleep disturbances and insomnia”. Please define “sleep disturbances” and “insomnia” before, if authors would like to describe them as differently in relation to pain.
  2. If authors would like to keep the heading of section 2 as “Preoperative sleep disturbances and insomnia” than separate out the following subsections where authors are describing about factors associating with postoperative sleep quality.
  3. How does childhood adversities, genetic vulnerability etc. are relevant in Figure 1? Please provide details somewhere.
  4. Section 2.2 first paragraph mentioned about thalamus…..Please explain its significance.
  5. Section 2.2 second paragraph, line “sleep disturbances……..factors” needs clarification.
  6. The authors described the role of inflammatory mediators, anesthesia and analgesia in sleep disturbances after surgery. However, it may little bit more meaningful if they could shed some light on temporal aspects. For example, how long does sleep problem persists after general anesthesia.
  7. In addition, how does anti-inflammatory medications post-surgery helps in preventing sleep problems.

Author Response

Thank you for the fast review of our MS entitled: “Sleep well and recover faster with less pain” and for the constructive comments.

Below, we answer each question (in bold italics). The changes in the revised manuscript are shown with “track changes”. In addition to the reviewers’ comments, we suggest that the title is revised to “Sleep well and recover faster with less pain – a narrative review on sleep in the perioperative period”

Looking forward to hearing from you,

Sincerely,

Reetta Sipilä, PhD

Review 2

This is an interesting review on benefits of sleep on postoperative pain recovery. However, the authors left some gaps and space between the information which made it difficult to grasp the topic most of the times. I would suggest that author revised this manuscript as suggested below to improve the comprehension.

1. The section 2 heading says “Preoperative sleep disturbances and insomnia”. Please define “sleep disturbances” and “insomnia” before, if authors would like to describe them as differently in relation to pain.

Thank you for this comment. We have already defined insomnia in the original manuscript (p. 2, line 62). In previous literature, sleep disturbances and insomnia are regularly used as synonyms. We decided to use here mainly the term “sleep disturbances”. This has now been changed in the text (section 2 heading, and the following paragraph (p2, line 62)

2. If authors would like to keep the heading of section 2 as “Preoperative sleep disturbances and insomnia” than separate out the following subsections where authors are describing about factors associating with postoperative sleep quality.

Thank you for this good suggestion. We have now replaced “preoperative” by “perioperative” which covers the whole period consisting of the pre-, per-, and postoperative periods.

3. How does childhood adversities, genetic vulnerability etc. are relevant in Figure 1? Please provide details somewhere.

Thank you for addressing this. The role of childhood adversities is explained in the text (section 2.1). We have now clarified this in the legend for Figure 1. Genes are mentioned in the Figure 1, since they have an important role in the regulation of all the variables in the ´vicious circle´.

4. Section 2.2 first paragraph mentioned about thalamus…..Please explain its significance.`

Thank you for addressing this. We agree that this would need a longer explanation of the role of thalamus in the  regulation of sleep. Therefore, we decided to remove this sentence from this review

5. Section 2.2 second paragraph, line “sleep disturbances……..factors” needs clarification. These factors are described in detail later under 2.3, 2.4., and 2.5. Thus, we have added “as described in more detail below” after “factors (p. 3, lines 115-116).

6. The authors described the role of inflammatory mediators, anesthesia and analgesia in sleep disturbances after surgery. However, it may little bit more meaningful if they could shed some light on temporal aspects. For example, how long does sleep problem persists after general anesthesia.

This is very interesting point. However, we could not find previous literature of the role of general anesthesia on persistence of sleep problems after surgery.

7. In addition, how does anti-inflammatory medications post-surgery helps in preventing sleep problems.                                                                                                                                This is a excellent question, thank you. We have now clarified this on p. 4, lines 141-144.

Reviewer 3 Report

The Authors, in their narrative review discussed the relevance of sleep disturbances both before and after surgery, how sleep disturbances should be assessed and monitored, and how better sleep can be supported by pharmacological and non-pharmacological interventions.

Although the matter is widely represented in literature, the Authors correctly approached and discussed this major clinical issue in anesthesiology.

The paper is well explained and clearly written, but some issues should be still addressed:

  1. As well as insomnia and respiratory sleep related disorders (mainly obstructive sleep apnea), the restless legs syndrome should be preoperatively investigated; RLS is an epidemiologically relevant disorder (2-3% in general population), underestimated, easily detectable (through medical history) and potentially misleading (the patient cannot fall asleep but he is not insomniac); moreover RLS symptoms are engendered or made worse by rest (lying or sitting); similarly, circadian sleep-wake rhythm disorders (mainly delayed and advanced sleep phase subjects) may be confounded with insomnia
  2. Sleep (and hypothetically general anesthesia) reduces the adenosine concentration in basal forebrain and consequently the sleep propensity (homeostatic system). In order to avoid sleep problems (falling asleep, sleep continuity), patients should be addressed to maintain the total sleep time about 7-8 h a day (preferably during the night, avoiding prolonged diurnal nap) already during preoperative evaluation (at home) and hospitalization
  3. As well as the homeostatic system, the circadian rhythm may be compromised during the hospitalization for many reasons (room lights, temperature, meal time, …); TV, computer, cells and tablets should be avoided during night since 10-11 pm (dim light melatonin onset); following a period of prolonged sleep (sedation), a correct circadian sleep-wake rhythm should be restored as soon as possible.
  4. Sleep hygiene rules should be eventually suggested to patients in preoperative evaluation and during hospitalization (see also 2 and 3)
  5. Given the bidirectional relationship between sleep and pain, an adequate control of pain remains a priority (I think that it should be addressed in 6. Summary and clinical implications).

Finally note that dexmedetomidina increases N2 sleep stage, but the glymphatic system is mainly activated during N3 sleep stage (slow wave sleep). In light of that, the paragraph “The noradrenergic tone has been shown to reduce NREM sleep and glymphatic flow, which has been associated with neuroinflammation since the glymphatic system has a pivotal role in removal of metabolic waste from the CNS [115]. Dexmedetomidine, being an a2-adrenergic agonist, may also have beneficial effects via this mechanism. In support of this hypothesis, Hu et al. [113] reported that the surgery-induced increase in IL-6 levels was lower in the dexmedetomidine-group.” (lines 308-314) should be revised.

Author Response

Thank you for the fast review of our MS entitled: “Sleep well and recover faster with less pain” and for the constructive comments.

Below, we answer each question (in bold italics). The changes in the revised manuscript are shown with “track changes”. In addition to the reviewers’ comments, we suggest that the title is revised to “Sleep well and recover faster with less pain – a narrative review on sleep in the perioperative period”

Looking forward to hearing from you,

Sincerely,

Reetta Sipilä, PhD

Review 3

The Authors, in their narrative review discussed the relevance of sleep disturbances both before and after surgery, how sleep disturbances should be assessed and monitored, and how better sleep can be supported by pharmacological and non-pharmacological interventions.

Although the matter is widely represented in literature, the Authors correctly approached and discussed this major clinical issue in anesthesiology.

The paper is well explained and clearly written, but some issues should be still addressed:

1. As well as insomnia and respiratory sleep related disorders (mainly obstructive sleep apnea), the restless legs syndrome should be preoperatively investigated; RLS is an epidemiologically relevant disorder (2-3% in general population), underestimated, easily detectable (through medical history) and potentially misleading (the patient cannot fall asleep but he is not insomniac); moreover RLS symptoms are engendered or made worse by rest (lying or sitting); similarly, circadian sleep-wake rhythm disorders (mainly delayed and advanced sleep phase subjects) may be confounded with insomnia.                                                Thank you for this important comment. We have now added text about this to p. 6, lines 252-256.

2-4. Sleep (and hypothetically general anesthesia) reduces the adenosine concentration in basal forebrain and consequently the sleep propensity (homeostatic system). In order to avoid sleep problems (falling asleep, sleep continuity), patients should be addressed to maintain the total sleep time about 7-8 h a day (preferably during the night, avoiding prolonged diurnal nap) already during preoperative evaluation (at home) and hospitalization.

As well as the homeostatic system, the circadian rhythm may be compromised during the hospitalization for many reasons (room lights, temperature, meal time, …); TV, computer, cells and tablets should be avoided during night since 10-11 pm (dim light melatonin onset); following a period of prolonged sleep (sedation), a correct circadian sleep-wake rhythm should be restored as soon as possible.

Sleep hygiene rules should be eventually suggested to patients in preoperative evaluation and during hospitalization (see also 2 and 3)

All of these (points 2-4) are excellent suggestions and we have now either added details about these topics in the text, or clarified our message (P.8, lines 281-286). Sleep hygiene rules and importance of adequate pain control are also now mentioned in Figure 2.

5. Given the bidirectional relationship between sleep and pain, an adequate control of pain remains a priority (I think that it should be addressed in 6. Summary and clinical implications).

The referee is absolutely right. The authors may have taken this message for granted. We have now added information about this into the text (p.10, line 370-371 and Figure 2).

6. Finally note that dexmedetomidina increases N2 sleep stage, but the glymphatic system is mainly activated during N3 sleep stage (slow wave sleep). In light of that, the paragraph “The noradrenergic tone has been shown to reduce NREM sleep and glymphatic flow, which has been associated with neuroinflammation since the glymphatic system has a pivotal role in removal of metabolic waste from the CNS [115]. Dexmedetomidine, being an a2-adrenergic agonist, may also have beneficial effects via this mechanism. In support of this hypothesis, Hu et al. [113] reported that the surgery-induced increase in IL-6 levels was lower in the dexmedetomidine-group.” (lines 308-314) should be revised.

Thank you for this interesting comment. However, we believe that what we have written is in accordance with what you write: dexmedetomidine increases N2 sleep (this is stated in the original MS, p. 9, line 312), we write that noradrenergic tone has been shown to reduce NREM sleep and glymphatic flow. This statement should be correct. We suggest that dexmedetomidine, being an a-2 adrenergic agonist could have beneficial effects on this. To clarify this, we have revised the text to: We suggest that dexmedetomidine, which blocks noradrenaline release from the locus coeruleus and has been shown to enhance glymphatic system transport in rodents, might have beneficial effects via this mechanism. We have added new references in support of this.

Round 2

Reviewer 2 Report

The authors have sufficiently addressed all my queries and concerns.